



# Northern Hemisphere atmospheric pattern enhancing Eastern Mediterranean Transient-type events during the past 1000 years

Aleix Cortina-Guerra[1], Juan José Gomez-Navarro[2], Belen Martrat[1], Juan Pedro Montávez[2], Alessandro Incarbona[3], Joan O. Grimalt[1], Marie-Alexandrine Sicre[4], P. Graham Mortyn[5,6]

[1]Department of Environmental Chemistry, Institute of Environmental Assessment and Water Research (IDAEA), Spanish Council for Scientific Research (CSIC), Barcelona, Spain.
[2]Department of Physics, University of Murcia, Murcia, Spain.
[3]Dipartimento di Scienze della Terra e del Mare, Università di Palermo, Palermo, Italy.
[4]Sorbonne Universités (UPMC, Université Paris 06)-CNRS-IRD-MNHN, LOCEAN Laboratory, Paris, France
[5]Institute of Environmental Science and Technology (ICTA), Universitat Autònoma de Barcelona, Bellaterra, Barcelona, Spain.
[6]Department of Geography, Universitat Autònoma de Barcelona, Bellaterra, Barcelona, Spain.

*Correspondence to*: Aleix Cortina (acortina@usal.es)

**Abstract.** High resolution climate model simulations for the last millennium were used to elucidate the main winter Northern Hemisphere atmospheric pattern during enhanced Eastern Mediterranean Transient (EMT-type) events, a situation in which an additional overturning cell is detected in the Mediterranean at the Aegean Sea. The differential upward heat flux between the Aegean Basin and the Gulf of Lions was taken as a proxy of EMT-type events and correlated with winter mean geopotential height at 500 mb in the Northern Hemisphere (200 N-900 N and 1000 W-800 E). Correlations revealed a pattern similar to the Eastern Atlantic / Western Russian (EA/WR) mode as the main driver of EMT-type events, with the past 1000 yr of EA/WR-like mode simulations being enhanced during insolation minima. Our model results are consistent with alkenone Sea Surface Temperature (SST) reconstructions that documented an increase in the west-east basin gradients during EMT-type events.

## 1 Introduction

The Mediterranean Sea is a small, semi-enclosed basin connected with the Atlantic Ocean through the Straits of Gibraltar (a 284 m deep sill at a width of ~30 km; (Bryden and Kinder, 1991)). The Sicily channel (average depth of 330 m, width of ~130 km; (Wüst, 1961)) subdivides the Mediterranean into a western and an eastern basin. An anti-estuarine pattern (Béthoux, 1979) characterizes the current Mediterranean general circulation, mainly driven by a negative water budget, involving the inflow of relatively fresh surface Atlantic waters and exit of relatively salty bottom Mediterranean waters (Fig. 1). The entering colder and fresher Atlantic waters interact with the warmer and saltier Mediterranean waters forming Modified Atlantic Water (MAW), which constitutes the main superficial water mass of the Mediterranean (0-200m)



(Malanotte-Rizzoli et al., 2014 and references therein). The MAW are the source of Levantine Intermediate Water (LIW; 200-600m), and both are involved in deep-water mass formation (Malanotte-Rizzoli et al., 2014). Northwesterly winds in the Adriatic Sea (Eastern Mediterranean Deep Waters, EMDW) and in the Gulf of Lions (Western Mediterranean Deep Water, WMDW) are key elements for enhanced deep-water ventilation (Millot, 1999).

An important perturbation in the Mediterranean overturning circulation took place in the late- 1980s to the mid-1990s that involved the formation of an additional overturning cell in the Aegean Sea (see Fig. 1) (Roether et al., 1996). This episode was termed the Eastern Mediterranean transient (EMT) event and involved major changes on the seawater physical and biogeochemical properties, including changes in the vertical and spatial distribution of anthropogenic carbon (Touratier and Goyet, 2011). Moreover, concurrent with the EMT event, a reduction of the Mixed Depth Layer (MDL) and Winter Heat

Flux in the Gulf of Lions (Beuvier et al., 2010; Herrmann et al., 2010) was observed, indicating a weakening of deep-water formation in the Western Mediterranean (Incarbona et al., 2016). Enhanced deep-water ventilation in the Eastern Mediterranean associated with wintertime cold polar/continental air outbreaks (Rohling et al., 2019) have been related with salinity minima in the Sicily channel both in recent (Gasparini et al., 2005) and past EMT-type events (Incarbona et al., 2016).


It has been suggested that the origin of EMT-type events could be related to modifications in atmospheric patterns operating at global scale such as the North Atlantic Oscillation (NAO) or East Atlantic / Western Russia (EA/WR) modes, low solar irradiance together with increase of volcanic eruptions (Incarbona et al., 2016). However, a robust demonstration using past climate model simulations is still lacking. Here, results for the past 1000 years of high-resolution (45 km) simulations carried

out with a Regional Climate Model (RCM) driven by a Global Circulation Model (GCM) are presented. This approach provides insight on how changes in global atmospheric circulation patterns affect Mediterranean heat loss, which are closely related to deep-water formation rates (Sur et al., 1993; Josey, 2003; Herrmann et al., 2010). The present study is, therefore, aimed to define the timing and the global atmospheric pattern of variability that enhanced EMT-type events.

## 2 Methods

### 2.1 Climate simulations

A GMC and a nested RCM have been used to produce a consistent climate simulation of the European climate for the past 1000 years. The GCM is the ECHO-G model, and consists of the spectral atmospheric model ECHAM4 coupled to the ocean model HOPE-G. This GCM setup has a spatial resolution of about 3.75° x 3.75° in the atmosphere and 2.8° x 2.8° in the ocean and has been employed and thoughtfully evaluated in the literature (Legutke et al., 2003). These data have been

dynamically downscaled with a RCM based on a climate version of the Fifth-Generation Pennsylvania-State University-National Center for Atmospheric Research Mesoscale Model (MM5). The model domain encompasses Europe and the





Mediterranean region entirely, and implements a spatial resolution of 45 km. As with ECHO-G, this model setup has been evaluated elsewhere (Gomez-Navarro et al., 2013; 2015). Both models are consistently driven by reconstructions of three external forcings: greenhouse gas concentrations in the atmosphere, long-term variations in Total Solar Irradiance (TSI) and

variations of Earth's orbital parameters. The results of coupling the RCM with the GCM are hereafter referred to as MM5-ECHO-G.

Upward heat fluxes calculated within MM5-ECHO-G are used in this study as a predictor of deep-water formation. In this regard, it is important to note that MM5-ECHO-G does not include a high-resolution regional ocean model. Instead, Sea Surface Temperature (SST) variations are directly taken from the driving GCM and imposed as an additional boundary

condition to the RCM. Still, the latter calculates the heat fluxes between the atmosphere and the surface, including the prescribed ocean SST, according to meteorological conditions. Therefore, the heat fluxes within the RCM simulation are consistently obtained according to the large-scale atmospheric circulation prescribed by the GCM, but improved according to the additional information provided by regional circulation features driven by the high-resolution orography and land mask of the RCM. Thus, monthly upward heat flux evaluation is needed to identify times of year when enhanced heat loss occurs.

MM5-ECHO-G simulations for the past 1000 years showed that on average, the September-February period (autumn-winter) (Fig. A1) accounted for most of the annual upward heat flux. Hence, the analysis is performed over this period, referred to as winter hereafter for convenience although it encompasses autumn months as well.

**2.2 Detection of the synoptic circulation pattern related heat flux variability**

In order to find the spatial pattern that most clearly influences deep-water formation in the Aegean Sea, we first define two

boxes delimiting deep-water formation areas in the Mediterranean: (1) Gulf of Lions (GL) ($41.5^0$ N – $43^0$ N, $3.5^0$ E - $6.5^0$ E) and (2) Aegean Basin (AB) ($35.7^0$ N - $37.5^0$ N, $23.5^0$ E -$27^0$ E) (Fig. 1). Winter upward heat flux difference between the AB and the GL is then calculated, so that positive values are associated with enhanced deep-water formation in the AB, and conversely with respect to the GL. As the magnitude of the upward heat flux in both regions can be very different and we seek relative variations, both series were standardized before estimating the difference. Thereby, we obtain the following

annual series:

$$\nabla_{HF}(t) = HF_{AB}(t) - HF_{GL}(t) \qquad\qquad (1)$$

where $\nabla_{HF}$ denotes the gradient of heat flux, and $HF_{AB}$ and $HF_{GL}$ the heat fluxes averaged for the aforementioned boxes and months after standardization. To find the spatial structure of atmospheric dynamic that most strongly affects the gradient of

upward heat flux, the series (eq:1) is correlated with the winter mean geopotential height at 500 mb (hereafter Z500) obtained from the driving GCM in the region $20^0$ N-$90^0$ N and $100^0$ W-$80^0$ E (Fig. 2):





$$\rho(x) = cor\big(\nabla_{HF}(t), Z500(x,t)\big) \qquad\qquad (2)$$

This pattern can be interpreted as a mode where associated variability is most strongly associated with the differences in deep-water formation between the AB and the GL.

Mathematically, this correlation map can be treated as a vector, and can be used to find an associated index by projecting the original Z500 field onto it. For this, the pattern has to be normalized first:


$$\rho(x) = \frac{\rho(x)}{\sqrt{\rho(x)\cdot\rho(x)}} \qquad\qquad (3)$$

where $\rho(x)$ represents the normalised vector and "·" is the scalar product. Now, the index that represents the "weight" of this pattern throughout the last millennium, but optimized for the explanation of deep-water formation in the AB, is simply
obtained as the projection of Z500 onto the pattern:

$$I'_{nhp}(t) = Z500(x,t) \cdot \rho(x) \qquad\qquad (4)$$

The variance of $I'_{nhp}$ (nhp stands for Northern Hemisphere pattern) can be compared to the total variance of the original
field of Z500, which results in 11% of the variance of the whole field. A possible drawback of the index defined by (4) is that it is affected by changes in global temperature, as geopotential height is closely related to temperature through the hypsometric equation. This implies that this index responds simultaneously to changes in atmospheric circulation, but also in global temperature. In order to overcome this problem keeping while the signal of the atmospheric dynamics, the spatially averaged Z500 is removed to define a new index. This is:


$$I_{nhp}(t) = Z500'(x,t) \cdot \rho(x) \qquad\qquad (5)$$

where $Z500'(x,t) = Z500(x,t) - \langle Z500(x,t)\rangle$ and "$\langle\rangle$" denotes spatial average.

Lastly, to complement this analysis and gain insight on the physical relationship between this circulation pattern and the
variables that modulate heat flux at the surface, we perform composite analysis based on the $I_{nhp}$ index. This analysis is carried out filtering out situations according to the aforementioned index values. In particular, dates corresponding to values over the 90[th] percentile are selected, and the corresponding fields of the variable target of the analysis are averaged. This is repeated for the dates of the lower 10[th] percentile values, and finally both averages are subtracted, yielding a map of anomalies that represents the impact of the index variability on the given variable. The rationale for this approach is that



under the null hypothesis of no relation whatsoever between the variability of the index to select dates and the variables, a composite is equivalent to a random selection of dates, which statistically cancels out after taking differences. And conversely, large deviations from zero, either positive or negative, are indicative of strong influence on the index on this variable.

## 2.3 Calculating western/eastern Sea Surface Temperature proxy

In order to validate model simulations a western/eastern alkenone-based SST gradient was calculated. Western (W) (Moreno et al., 2012; Nieto-Moreno et al., 2013; Sicre et al., 2016) and eastern (E) (Versteegh et al., 2007; Grauel et al., 2013; Gogou et al., 2016) marine SST proxies were first standardized (Table S1) and average values of a period before, during and after solar minimum events (Crowley et al., 2000) of both basins were calculated in order to evaluate the evolution of W-E gradients around solar minima (Table 1). The length of the period chosen to calculate average SST values was equal to the
duration of solar minimum.

## 3 Results and discussion

### 3.1 Identification of the Northern Hemisphere atmospheric pattern most closely related to EMT-type events

The correlation coefficient between the AB-GL gradient and Z500' in the Northern Hemisphere (Fig. 2) reveals a pattern characterized by positive correlations located over Europe, and flanked by negative correlations over the central North
Atlantic and over Western Russia. This pattern is reminiscent of the EA/WR pattern defined by the NOAA Climate Prediction Center (CPC), although the latter is obtained through rotated principal component analysis (Barnston and Livezey, 1987) of the observed monthly mean 500 mb height anomaly field in the region 20ºN-90ºN. The impacts on air-sea heat exchange of variability modes affecting the Mediterranean have been studied by Josey et al. (2011). To relate air-sea heat exchange, deep-water formation and atmospheric circulation, Josey et al. (2011) used a top-bottom approach consisting
of decomposing atmospheric dynamics in their more prominent modes of variability and associated indices, and then looking for relationships between such modes and surface heat flux release in the Mediterranean. The results of this analysis revealed that the EA/WR mode most likely plays a major role in the deep-water formation in the AB. In our study, we applied a different strategy by undertaking a bottom-up approach, where the phenomenon to explain, i.e. changes in the locations of deep-water formation in the Mediterranean, is used to find a pattern based on physical processes. This type of approach
enables more flexibility, as it allows the associated index to be optimized to explain the fraction of the atmospheric variability that most directly affects the given phenomenon, hence maximizing the signal sought. Therefore, the fact that the pattern obtained through a completely different approach resembles the EA/WR structure reinforces the findings of Josey et al. (2011) and extends them over the longer temporal frame of the past 1000 years. Our results demonstrate that the index representing the "weight" of this correlation pattern through the last millennium, calculated in equations (4) and (5) (i.e. $I_{nhp}$),



can be used as a proxy of EA/WR-like variability. This variability is associated with changes in the deep water formation zones and, in particular, to the occurrence of EMT-type events.

### 3.2 Heat loss in Mediterranean Sea during EMT-type events

To gain insight on how the EA/WR variability mode is related to changes in heat exchange, we have obtained composites of various variables defined according to the $I_{nhp}$ index. To calculate the net heat exchange between sea and atmosphere, four

components should be taken into consideration: (1) sensible heat flux, (2) latent heat flux, (3) longwave flux and (4) shortwave flux. Winter net heat exchange is dominated by latent heat flux and to a lesser extent by sensible heat flux (Josey, 2003). These two components are driven by the product of the wind speed and the sea-air humidity and the sea-air temperature gradient (Josey et al., 1999). Therefore, to unravel the driving mechanisms of sea surface heat loss associated with the EA/WR-like mode, it is necessary to consider anomalous wind speed and air temperature fields (the atmospheric

humidity field tends to follow air temperature and it is neglected) (Josey et al., 2011).

The composites of winter 2-m air temperature (i.e. near surface air temperature), upward heat flux and 10-m wind speed, obtained using the $I_{nhp}$ index, are shown in Fig. 3. The intensification of the spatial pattern described in the former section is associated with an increased western flux in the eastern Mediterranean, which favors the intensification of cold winds from

the continental regions that, in turn, increase the upward heat flux in this region promoting deep-water formation. Conversely, the pattern tends to reduce this zonal flow over the western Mediterranean, which therefore reduces the heat flux exchange there. These changes are summarized in the heat anomaly pattern of the top panel of Fig. 3, which is associated with an increased gradient between the AB and GL. This can also be appreciated in the near surface temperature pattern, with the warm (cold) anomaly in the western (eastern) Mediterranean driven by reduced (enhanced) zonal flow, and that

agrees with the anomalies of heat exchange aforementioned. This pattern is due to the anomalous high-pressure system centered over the North Sea that results in cold northwesterly airflow over the eastern Mediterranean and Black Sea, and a warmer southeasterly airflow in the western Mediterranean, generating a dipole in the heat exchange (Josey et al., 2011). Usually in the Mediterranean Sea, the Levantine basin is characterized by higher temperatures, and high differences in the Evaporation-Precipitation balance facilitates LIW formation (Millot, 1999). Considering near surface air temperature varying

in parallel with SST the predominance of this mode of variability results in reducing or compensating the average temperature gradient in the Mediterranean.

### 3.3 EA/WR-like pattern variability during the past 1000 years and its influence on Mediterranean climate

Solar activity and last millennium EA/WR-like pattern variability $I_{nhp}$ are shown in Fig. 4a-b. After applying eq(5) to $I'_{nhp}$, the global temperature signal, and thus the possible thermodynamic effect of solar forcing on the index, was removed. The

residual signal is solely attributed to variations in the atmospheric circulation. When comparing the $I_{nhp}$ variability with solar forcing (Crowley, 2000), a good correspondence is revealed for the analyzed interval. In particular, the Lomb periodogram





(Fig. 4f) reveals significant peaks of both signals with a ~125 yr periodicity (frequency = 0.008 yr$^{-1}$). After applying a Gaussian filter to both signals (Fig. 4e), frequency=0.008 ± 0.001 yr$^{-1}$ (i.e. 110-140 yr periodicity range), a strong relationship arises (r=-0.83, p<0.001). Interestingly, a similar variability has been previously documented (Baumgartner et

al., 1992; Patterson et al., 2004, 2005; Cortina and Herguera, 2014, among others), attributed to solar activity expressed as changes in the $^{14}$C content of the atmosphere($\Delta^{14}$C) (Neftel et al., 1981; Sonett, 1984; Stuiver and Braziunas, 1993). Our analysis suggests that solar activity minima with approximately 125 yr periodicity is related to $I_{nhp}$ enhancement and the ensuing expression of EA/WR-like atmospheric patterns. The latter is related to generation of EMT-type events through the physical relationship described above. Our results are in line with previous interpretations of circulation perturbation in the

Mediterranean by Incarbona et al. (2016) who related solar irradiance lows with enhancement of EMT-type events, but we restrict this relationship to a 125 yr cycle. The length of the simulation (1000 yr) could preclude detection of longer periodicities, and low resolution of the solar forcing proxy from year 1000 to 1700 (Crowley et al., 2000) could prevent evaluation of low periodicities such as the 88-yr Gleissberg cycle (Gleissberg and Schove, 1958). In fact, a 90-80-yr periodicity is present in the $I_{nhp}$ index (Fig. 4f), and could be responsible for the last EMT event.


Model simulations were also compared with oceanic proxy reconstructions during three singular periods of solar minima: (1) Maunder (1645-1715 yr), (2) Dalton (1790-1830 yr) and (3) Gleissberg (1900-1920 yr) (Table 1)(Fig. 4c, 4d). Since EMT-type events co-occurred with freshening events in the Sicily channel (Incarbona et al., 2016), anomalous low $\delta^{18}$O seawater values in this region (Fig. 4d) should be contemporaneous with enhanced $I_{nhp}$ associated with EA/WR-like mode. This

correspondence is precise during the Gleissberg and Dalton minima, and the 10-year lag observed between the freshening event and the end of Maunder minimum (i.e. 1715 AD) is within its own chronological uncertainty (±25 years; (Incarbona et al., 2016)). On the other hand, the near surface temperature composite map revealed a reduced or compensated average temperature gradient between western and eastern Mediterranean basins during enhanced $I_{nhp}$ (EA/WR-like) pattern (Fig. 3). The W-E gradient derived from SST proxy reconstructions (Fig. 4c), that is independent from model simulations, agrees

with these results, showing higher values (i.e. increased difference between western and eastern basin SSTs) during solar minima and an enhanced $I_{nhp}$ (EA/WR-like pattern).

The fact that the EA/WR-like mode dominated periods with increased differential upward heat flux between AB and GL, increased W-E temperature gradient and hence the occurrence of EMT-type events, does not exclude the influence of other

important modes of atmospheric variability, such as positive phases of the North Atlantic Oscillation (NAO) (Incarbona et al., 2016). The EA/WR-like pattern explains about 11% of atmospheric variability in the simulation, whereas studies based on Principal Component Analysis suggest that NAO accounts for about 40% of total variance, demonstrating the strong influence of this mode on North Atlantic atmospheric circulation. However, our model simulation results discard a direct influence of positive NAO during periods with an increased upward heat flux gradient between AB and GL, restricting its

impact most likely to atmospheric preconditioning.



## 4 Conclusions

The MM5-ECHO-G simulation can be used to characterize the global EA/WR-like atmospheric mode in the Mediterranean region, which favors continental cold winds to penetrate into the AB, and blocks their influence in the GL. The model results predict an increase in the winter upward heat flux gradient between the AB and GL, enhanced Mediterranean deep-water

formation in the eastern basin, with its impact on the circulation of the entire basin. At present, these oceanographic conditions have been related to the EMT event, which demonstrates the suitability of this model configuration to study the variability of EMT-like events in the past. Our results show that during the past 1000 yr, a dominant EA/WR-like mode and EMT-type events, were contemporaneous with solar minima, likely related with cycles of approximately 125 and 80-90 years.


The model simulation is consistent with the multi-decadal return period of surface freshening in the Sicily channel, a proxy of EMT-type events, for the Maunder (1645-1715 yr), (2) Dalton (1790-1830 yr) and (3) Gleissberg (1900-1920 yr) minima. Moreover, the simulation results are in line with alkenone-based SST proxies that document an increase of the W-E gradient during these periods as consequence of winter-time northerly air outbreaks over the AB.

**Author contribution**

AC and JJGN developed the methodology, performed the format analysis and prepared the original draft

JPM and JOG reviewed and edited the manuscript

BM participated in the funding acquisition and conceptualization

AI, MAS and PGM participated in the conceptualization

**Acknowledgements**

This work started as a collaboration between researchers with the PALEOLINK project by the PAGES 2k Network. We acknowledge support from the PAGES (Past Global Changes) 2k Network, funded by the U.S. and Swiss National Science Foundations (NSF) and NOAA. We want also to acknowledge project PGC2018-102288-B-I00 founded by Ministerio de Ciencia, Innovación y Universidades. J.J.G.N. acknowledges the funding obtained through the "Juan de la Cierva-

Incorporación" program (IJCI-2015-26914).




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






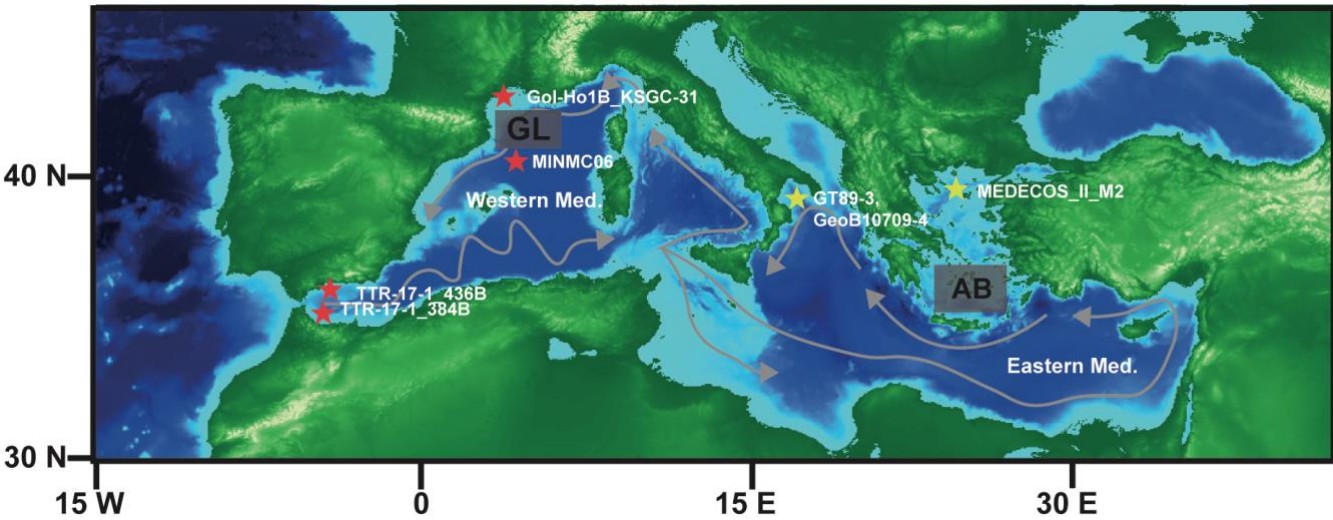

**Figure 1: Map of the Mediterranean Sea modified after Incarbona et al. (2016). Grey arrows depict main surface water paths. Stars show the location of cores used to calculate the Sea Surface Temperature (SST) gradient between western (red stars) and eastern (yellow stars) basins (W-E). Shaded rectangles show the area taken for estimation of the differential winter upward heat flux between the Aegean Basin and Gulf of Lions (AB-GL).**


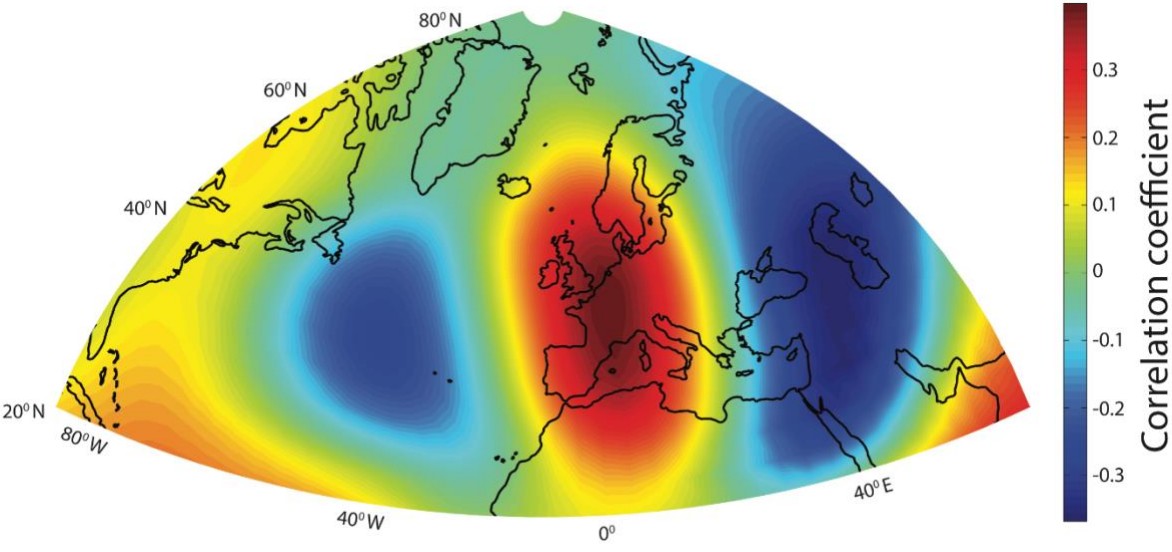

**Figure 2: Correlation map between winter upward heat flux gradient (Aegean Basin versus Gulf of Lions difference) and winter mean geopotential height at 500 mb (Z500'). The map reveals the Northern Hemisphere atmospheric pattern most closely related to EMT-type events.**


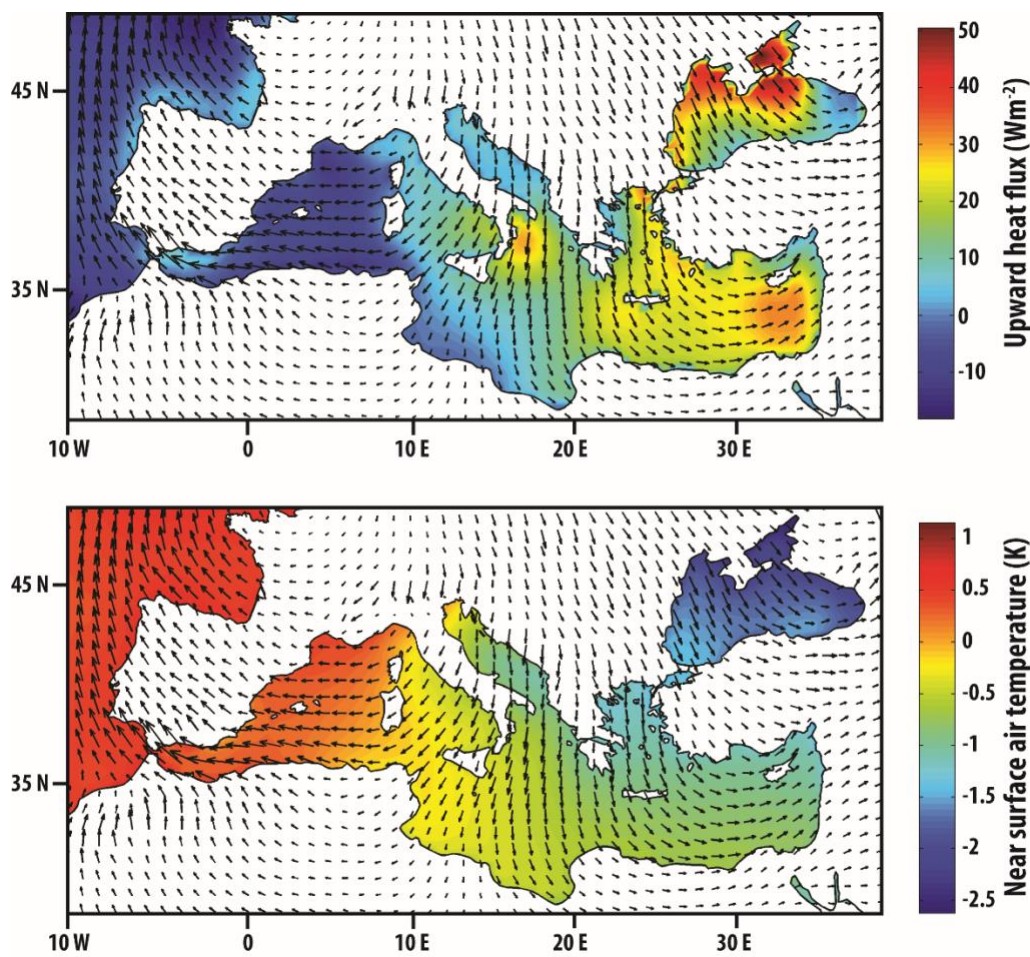


**Figure 3: Mediterranean composite maps.** Composite maps of winter near surface air temperature (2 m) and upward heat flux based on $I_{nhp}$ index. Dates corresponding to values over the 90th percentile are selected, and the corresponding fields of the variable target of the analysis are averaged. This is repeated for the dates of the lower 10th percentile values, and finally both averages are subtracted. Black arrows represent composite of 10-m wind speed direction and intensity.

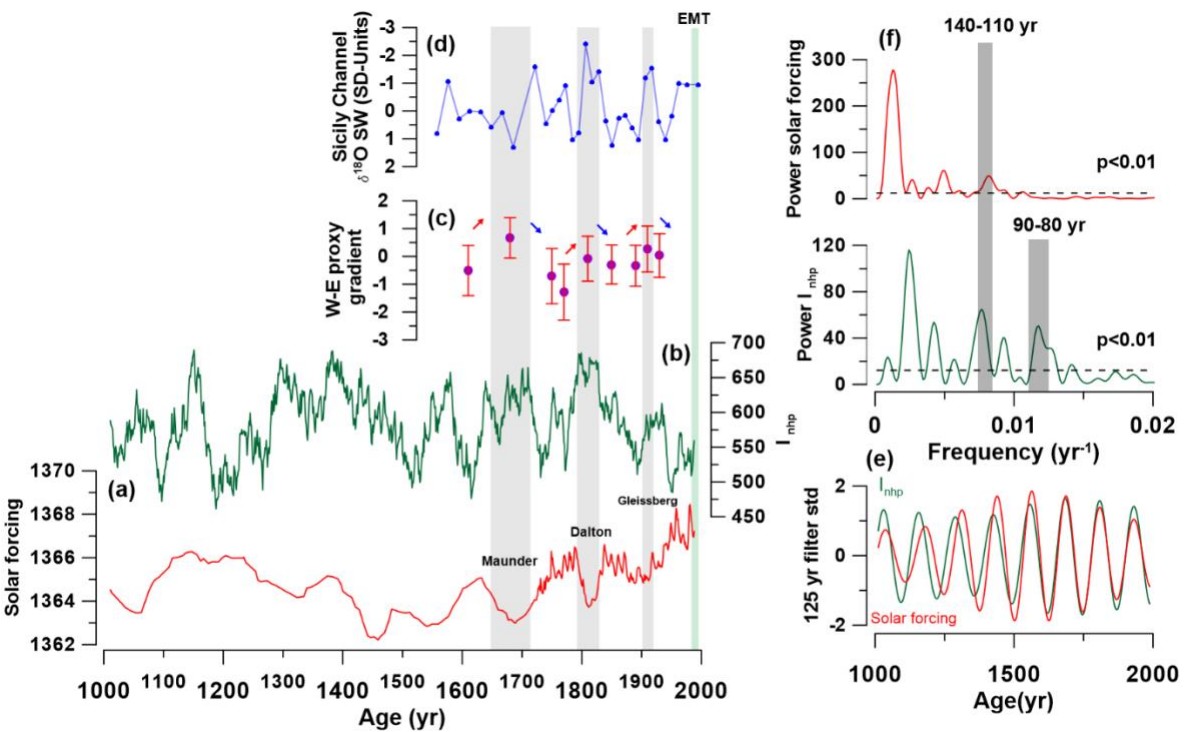


**Figure 4. Past EA/WR-like pattern variability and its correspondence with climate proxies. (a) Solar forcing (Crowley, 2000). (b) $I_{nhp}$ index. (c) Western-eastern basin SST gradient. (d) Sicily channel $\delta^{18}O$ sea water in standard units (Incarbona et al., 2016) (e) Standardized Gaussian filter centered at 0.008 yr-1 (i.e. 125 yr) with 0.001 yr-1 bandwidth (i.e. 110 yr-140 yr) of $I_{nhp}$ index and Solar forcing. (f) Power spectra of solar forcing and $I_{nhp}$ index based on Lomb periodogram algorithm using the PAST 3.12 software package (Hammer et al., 2001). Dashed lines representing white noise (p<0.01). Grey bars at figures a – d represent periods of solar irradiance lows. Green bar in panels a-d refer to the last EMT event.**







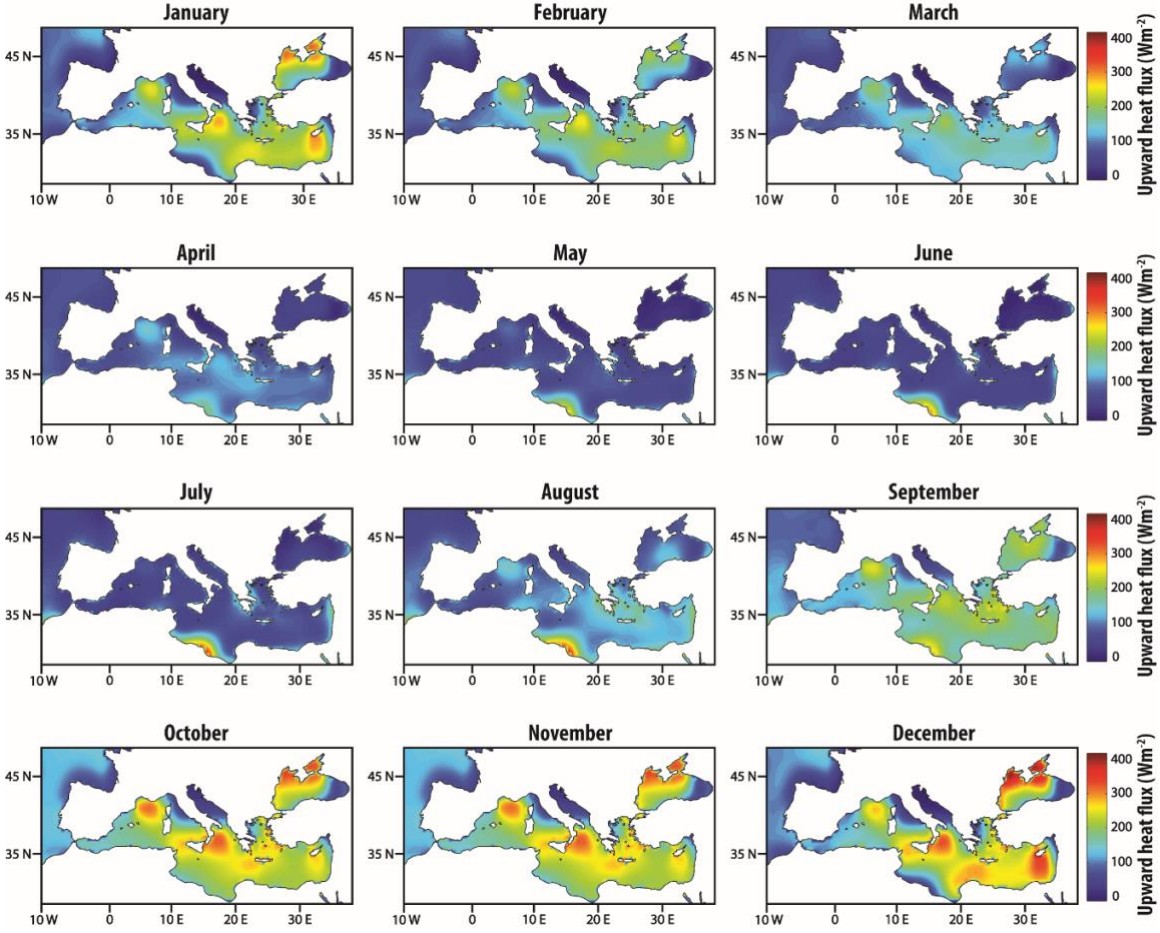

**Figure A1. Monthly upward heat flux for MM5-ECHO-G paleosimulations for the last 1000 years identify September-February interval (autumn-winter) as the period contributing to most of the annual upward heat flux. Air-sea exchanges during this winter-centered half of the year spans the main period for deep water formation. In this study, for convenience this period is referred to as winter (although it contains the outlying months of September and October which lie outside of a typical winter).**







| Period | Mean W | std W | n W | Mean E | std E | n E | W-E | std W-E |
|---|---|---|---|---|---|---|---|---|
| 1920 - 1940 | -0.15 | 0.51 | 4 | -0.18 | 0.83 | 15 | 0.03 | 0.78 |
| 1900 - 1920* | -0.10 | 1.17 | 3 | -0.37 | 0.75 | 13 | 0.27 | 0.82 |
| 1880 - 1900 | -0.15 | 1.00 | 5 | 0.19 | 0.64 | 15 | -0.34 | 0.73 |
| 1830 - 1870 | -0.65 | 0.42 | 9 | -0.36 | 0.76 | 29 | -0.29 | 0.70 |
| 1790 - 1830* | 0.06 | 0.98 | 7 | 0.15 | 0.75 | 22 | -0.08 | 0.80 |
| 1750 - 1790 | -0.57 | 0.47 | 8 | 0.71 | 1.11 | 27 | -1.28 | 1.01 |
| 1715 - 1785 | -0.72 | 0.51 | 14 | -0.02 | 1.09 | 46 | -0.71 | 0.99 |
| 1645 - 1715* | -0.15 | 0.88 | 19 | -0.81 | 0.61 | 29 | 0.67 | 0.73 |
| 1575 - 1645 | -0.36 | 0.69 | 15 | 0.15 | 1.01 | 23 | -0.51 | 0.90 |

**Table 1: Periods used to calculate western/eastern alkenone-based Sea Surface Temperature (SST) gradient (W-E). Mean,**
**standard deviation (std) and number of cases (n) is supplied. * Denotes periods corresponding to solar minima**