# Peer review of "Northern Hemisphere atmospheric pattern enhancing Eastern Mediterranean Transient-type events during the past 1000 years"

_Climate of the Past, 2021_

## Author Response (AR1)

REVIEWER 1

The manuscript is very well presented; spotting light an interesting issue: Eastern Mediterranean Transient pattern and development over a considerable range of time. The methodology is clear and easy to follow. Results are comprehensive. Only tiny technical (writing) corrections are needed prior to publication:

1. Authors are advised to use the symbol ° or the superscribed O instead of the superscribed 0. Please review the abstract (it is 20° NOT 200....etc) and Page 3 L 86 and L 96.

R. We have changed symbols as suggested by the reviewer

2. In Equation (1) I suggest using the symbol Δ to express the difference instead of ∇, which may mislead to the nabla of the partial differential vectors.

R. We have changed symbols as suggested by the reviewer

3. Reference Legutke et al. (2003) should be corrected to Zorita et al. (2003) and re-written correctly in the reference list.
R. We have changed reference as suggested by the reviewer

4. Either change the reference Crowley et al. (2000) in the manuscript to Crowley (2000) OR add it to the reference list.

R. We have changed reference to Crowley (2000)

REVIEWER 2

Review of the Manuscript cp-20021-24 Northern Hemisphere atmospheric pattern enhancing Eastern Mediterranean Transient-type events during the past 1000 years" by Cortina-Guerra et al.

General comment to the Authors and the Editor:

The ms presents an analysis, based on high resolution climate model simulations for the last 1000 years, of the main atmospheric patterns that might have driven past EMT-like events, which the Aegean Sea acting as an additional (or substitute) deep water mass formation area for the eastern Mediterranean Sea.

The ms is well organized, clearly written, with a logical structure that guides the reader through the author's reasoning. However, you should also say that for the identification of the past EMT-like events, you rely on a strong assumption, i.e. that every time an EMT event occurred this is seen in the difference of heat fluxes between the GoL and the Aegean. Of course, you cannot know that this is always the signature, and you cannot know that strong differences might occur, even if no EMT event occur. This is the basic assumption you do, and you should be clear about its limitations. In

addition, I found that you did not mention other studies that have for instance reported an EMT-like event during the 70s, which does not appear in your study. You should mention this discrepancy, and if possible explain it (some references are at the end, see *).

R. We agree with the reviewer that we have to state the limitation about considering an EMT event only if this strong difference occurs in deep water formation between the two basins, so we added a sentence (L 84-85). About the EMT in the 70s, it appears in our dataset, we add a sentence explaining that is detected by the $I_{nhp}$ index (L 204-206)

I recommend publication of the ms after moderate revision

Some more detailed comments are:

L28 should be Strait, not Straits
R. We have changed as suggested by the reviewer

L29 why for the Sicily channel you indicate an average depth, and not the sill depth (which is about 500m), as you did for Gibraltar?
R. We have changed as to sill depth.

L34 the term MAW is no longer used, given that all Mediterranean water masses are a modification product of the water coming from the Atlantic. Use simply AW and rewrite the sentence accordingly
R. We have changed as suggested by the reviewer
L34 should be surface, instead of surficial
R. We have changed as suggested by the reviewer

L35 "AW is the source"
R. We have changed as suggested by the reviewer

L37 should be "where Eastern Mediterranean Deep Water, EMDW, forms" and "where Western Mediterranean Deep Water, WMDW, forms". It is "Gulf of Lion" not "Lions". At the end of the sentence, add "through deep convection" for more clarity
R. We have changed as suggested by the reviewer

L41 "additional" or substitute, since it mostly replaced the Aegean?
R. We deleted. "An important perturbation in the Mediterranean overturning circulation took place in the late- 1980s to the mid-1990s that involved the formation of an overturning cell in the Aegean Sea"

L42 should be "Transient"
R. We have changed as suggested by the reviewer

L44 should be "Mixed Layer Depth (MLD)", not MDL
R. We have changed as suggested by the reviewer
L44-45 do not capitalize Winter Heat Flux

R. We have changed as suggested by the reviewer

L48 salinity minima at which layer? Please specify
R. Changed by sea surface salinity minima.
L58 announce the aim more clearer, by saying that it is to identify past EMT-type events and to define the timing and the global atmospheric….
R. Sentence rewroted

L93 write "the standardized heat fluxes…", and remove "after standardization" in L94
R. We have changed as suggested by the reviewer

L94 which months?
R. Added months although it is expressed above when we say all the analysis is performed for these month (L-80)

L139-140 it is not clear to me, what this sentence should say in relation to the EMT like events
R. This lines are to stablish that the box before and after the minimum solar used to calculate average SST has the same length than the time for Maunder, Dalton, y Gleissberg minima.

L165 replace "should" with "need"
R. We have changed as suggested by the reviewer

L221 "studies based on PCA"….you should give the references to these studies
We will references about this statement.
R. We added reference, Jianping and Wang, 2003

L226-239 Conclusions are too short

R. We added the limitation about considering an EMT event only occurs when a strong difference in heat fluxes between AB and GL is assessed.

L229 is it correct to use the verb "predict" while speaking about past events=
R. We changed by assessed

L230 replace "eastern basin" with "Aegean" because also the Adriatic is in the eastern basin, but I guess you are talking about the EMT like events.

R. We have changed as suggested by the reviewer

Figure 1 the names located near the stars, are not mentioned anywhere in the text
RThese are the names of the cores selected in order to calculate averaged SST in both basins. In order to make more clear these calculations we are going to add Sup material with this data.

Figure 4, the authors should briefly evidence and explain the fact the EMT during the 80s-90s is the shortest one they detected
R. We are going to clarify this fact on the main text following the main suggestion of the reviewer about EMT in the 70's. (L 204-206)

* Several papers [Lascaratos et al., 1999; Skliris and Lascaratos, 2004; Skliris et al., 2007; Beuvier et al., 2010; Vervatis et al., 2013; Theocharis et al., 2014] based on both observations and numerical modeling have reported a similar event taking place in the eastern part of the EMed during the 1970s.

Lascaratos, A., W. Roether, K., Nittis, and B. Klein (1999), Recent changes in the deep water formation and spreading in the Eastern Mediterranean Sea, Prog. Oceanogr., 44(1–3), 5–36.

Skliris, N., and A. Lascaratos (2004), Impacts of the Nile River damming on the thermohaline circulation and water mass characteristics of the Mediterranean Sea, J. Mar. Syst., 52, 121–143, doi:10.1016/j.jmarsys.2004.02.005.

Skliris, N., S. Sofianos, and A. Lascaratos (2007), Hydrological changes in the Mediterranean Sea in relation to changes in the freshwater budget: A numerical modelling study, J. Mar. Syst., 65, 400–416, doi:10.1016/j.jmarsys.2006.01.015.

Vervatis, V. D., S. S. Sofianos, N. Skliris, S. Somot, A. Lascaratos, and M. Rixen (2013), Mechanisms controlling the thermohaline circulation pattern variability in the Aegean–Levantine region, A hindcast simulation (1960–2000) with an eddy resolving model, Deep Sea Res., Part I, 74, 82–97, doi:10.1016/j.dsr.2012.12.011

Theocharis, A., G. Krokos, D. Velaoras and G. Korres (2014), An internal mechanism driving the alternation of the Eastern Mediterranean dense/deep water sources, In The Mediterranean Sea: Temporal Variability and Spatial Patterns, edited by G. L. E. Borzelli, et al., AGU Geophys. Monogr. Ser., 202, pp. 113–137, John Wiley, Oxford, U. K., doi:10.1002/9781118847572.ch8.

Beuvier, J., F. Sevault, M. Herrmann, H. Kontoyiannis, W. Ludwig, M. Rixen, E. Stanev, K. Béranger, and S. Somot (2010), Modeling the Mediterranean Sea interannual variability during 1961–2000: Focus on the Eastern Mediterranean Transient, J. Geophys. Res., 115, C08017, doi:10.1029/2009JC005950.